# Ablation of *SMUG1* Reduces Cell Viability and Increases UVC-Mediated Apoptosis in Hepatocarcinoma HepG2 Cells

**DOI:** 10.3390/genes12020201

**Published:** 2021-01-30

**Authors:** Mi-Jin An, Geun-Seup Shin, Hyun-Min Lee, Jung-Woong Kim

**Affiliations:** Department of Life Science, Chung-Ang University, Seoul 06974, Korea; dksalwls333@gmail.com (M.-J.A.); rjstjq89@naver.com (G.-S.S.); lhmscb@naver.com (H.-M.L.)

**Keywords:** *SMUG1*, knock-out, RNA-seq, proliferation, DNA damage repair

## Abstract

Uracil is an unavoidable aberrant base in DNA sequences, the repair of which takes place by a highly efficient base excision repair mechanism. The removal of uracil from the genome requires multiple biochemical steps with conformational changes of DNA that inhibit DNA replication and interfere with transcription. However, the relevance of uracil in DNA for cellular physiology and transcriptional regulation is not fully understood. We investigated the functional roles of *SMUG1* using knock-down (KD) and knock-out (KO) models. The proliferation ratio of *SMUG1* KD and KO cells was decreased compared to WT control cells, and the cell cycle was arrested in the G2/M phases before the transition to mitosis. The apoptotic cell death was increased in KD and KO cell lines through the increase of BAX and active caspase 3 expression. Phospho-gamma-H2AX expression, which reflected accumulated DNA damage, was also increased in KO cells. Moreover, the apoptotic cells by DNA damage accumulation were markedly increased in *SMUG1* KD and KO cells after ultraviolet C irradiation. Transcriptomic analysis using RNA-seq revealed that *SMUG1* was involved in gene sets expression including cell cycle transition and chromatin silencing. Together, the results implicate SMUG1 as a critical factor in cell cycle and transcriptional regulation.

## 1. Introduction

In aerobic organisms, such as animals, fungi, and bacteria, reactive oxygen species (ROS) are produced by normal metabolism and cause oxidative damage to DNA [1]. Base excision repair (BER) is a highly conserved mechanism from bacteria to humans, and it is essential for repairing endogenous DNA damage, including that caused by alkylation, oxidation, deamination, depurination, and single-strand breaks [2]. Hypoxia suppresses DNA repair pathways that contribute to genomic instability, and this results in intracellular accumulation of DNA damage [3,4,5,6]. In addition, ultraviolet (UV)-induced ROS act as powerful oxidants that may cause oxidative DNA damage [7]. The UV-induced DNA lesions, if not repaired, may cause severe structural distortions in the DNA molecule, thereby affecting important cellular processes, such as DNA replication and transcription, compromising cellular viability and functional integrity and ultimately leading to mutagenesis, tumorigenesis, and cell death [8,9].

In the mammalian cell, four different uracil-DNA glycosylase (*UDG*) genes have been identified, including *SMUG1*, *UNG*, *TDG*, and *MBD4*. The UDG family is a family of enzymes that are crucial to DNA repair; their absence allows some mutations to cause cancer [10]. Current evidence suggests that, in human cells, TDG and SMUG1 are the major enzymes responsible for the repair of the U:G mismatch caused by spontaneous cytosine deamination, whereas uracil arising in DNA through dUTP misincorporation is mainly dealt with by UNG [11]. The *SMUG1* (single-strand selective monofunctional uracil-DNA glycosylase) gene plays roles in various molecular functions, such as DNA binding, DNA N-glycosylase activity, and single-strand selective and uracil-DNA N-glycosylase activity. Additionally, SMUG1 is also a key enzyme for repairing 5-hydroxymethyluracil, 5-formyluracil, 5,6-dihydrouracil, alloxan, and other lesions generated during oxidative base damage induced by ionizing radiation and oxygen free radicals [12]. Although SMUG1 is involved in DNA repair in damaged cells, the functional role of SMUG1 in recognizing the damaged DNA regions in the genome and repairing mechanisms remains unclear.

To understand the role of SMUG1 in the regulation of DNA damage responses, we generated knock-down (KD) and knock-out (KO) *SMUG1* cell lines using the CRISPR-Cas9 gene-editing system. *SMUG1* KO reduces cell proliferation and induces apoptosis. In addition, *SMUG1* KD and KO cells were hyposensitive to DNA damage caused by ultraviolet C (UVC) irradiation, and *SMUG1* ablation led to apoptosis by delaying the cell cycle. Transcriptome analysis newly revealed that SMUG1 is involved in cell cycle transition and chromatin organization. These results highlight the involvement of SMUG1 in the regulation of DNA damage responses.

## 2. Materials and Methods

### 2.1. Cell Cultures and Transfection

All the cell culture reagents used in this study were purchased from Welgene (Seoul, Korea). HepG2 and HEK293T cells were purchased from the Korean Cell Line Bank (Seoul, Korea). Cells were maintained in Dulbecco’s modified Eagle medium (DMEM) supplemented with 10% fetal bovine serum (FBS) and 1% penicillin/streptomycin at 37 °C in an incubator with 5% CO_2_ in a humidified atmosphere. For the experiments, the coverslips were treated with 0.1 mg/mL of poly-D lysine for 6 h at room temperature, and they were placed in 100 mm cell culture dishes. Cells were seeded at a density of 2.5 × 10^6^ cells per well in 100 mm cell culture dishes. The other cells in 100 mm cell culture dishes were gently washed with Dulbecco’s phosphate-buffered saline without calcium and magnesium, and then cells were trypsinized. Transient transfection was performed by lipofectamine (Invitrogen, Carlsbad, CA, USA) with different plasmid DNA according to the manufacturer’s instructions.

### 2.2. Cell Cycle Analysis by the Flowcytometry

Cell cycle analysis was performed using propidium iodide (PI) staining (Sigma, Burlington, MA, USA). Cells were dissociated with trypsin-EDTA (Welgene, Seoul, Korea) and resuspended with 300 mL of PBS and fixed with 70% ethanol at 4 °C for 1 h. Cell pellets were then resuspended in 0.2 mL of PBS containing 0.25 μg/μL RNase A for 1 h at 37 °C. After that, cells were stained with 10 mL of propidium iodide (PI) solution (1 mg/mL) at room temperature in a dark condition. Finally, 1 × PBS was added to the PI-stained cells and was analyzed by BD Accuri™ C6 Plus (BD FACS, San Jose, CA, USA). At least 10,000 cells were used for each analysis, and the results were displayed as histograms. To investigate S phase progression, we used a double-thymidine block to synchronize cells at the G1 stage and release for the indicated time (0, 1, 2, and 10 h). After releasing, the cells were incubated with EdU for 2 h and stained with both PI and iFluor 488 azide dye. The percentage of cell distribution in the Sub-G1, G0/G1, S, and G2/M phases was measured, and the results were analyzed by the BD Accuri™ C6 Plus software for cell cycle profile.

### 2.3. Apoptosis Analysis

Cell apoptosis analysis was performed by using FITC Annexin-V Apoptosis Detection Kit I (556547; BD Sceince, Franklin Lakes, NJ, USA). Cells were trypsinized and centrifuged at 3000 rpm for 3 min, and the pellets were washed with ice-cold PBS. Then, cells were resuspended with 100 µL of 1 × binding buffer and stained with 5 mL of PI, FITC-Annexin V for 15 min in the dark condition. After PI and FITC-Annexin V staining, cells were mixed with 400 μL of 1 × binding buffer and analyzed by flow cytometry.

### 2.4. Generation of SMUG1 Knock-Out (KO) and Knock-Down (KD) Stable Cell Lines

The SMUG1 sgRNA target sequence was designed using the sgRNA prediction program (http://crispor.tefor.net). Each oligo was phosphorylated and annealed using T4 polynucleotide kinase (NEB, Ipswich, MA, USA). The lenti-sgRNA puro vector (Addgene, #104990) was digested with BsmBI (NEB, Ipswich, MA, USA) at 55 °C for 1 h. The digested vector was separated by gel electrophoresis and extracted using a Gel/PCR purification kit (Favorgen, Ping-Tung, Taiwan). The digested vector was checked with Nanodrop to confirm good DNA quality. The annealed oligo was ligated into the BsmBI-digested lenti-sgRNA puro vector at 4 °C overnight. For the transformation of bacteria, ligation reaction was mixed with 100 mL competent DH5a (RBC, #RH617). Sequencing of cloned lenti-sgRNA puro was performed using U6 promoter sequence (5′-GGG CAG GAA GAG GGC CTA T-3′) by Cosmo genetech. The construct was transfected to HepG2 cells and treated with 1 mg/ mL of puromycin for 4 to 7 days. Selected cells were recovered with Dulbecco’s modified Eagle medium (DMEM) supplemented with 10% fetal bovine serum (FBS) and 1% penicillin/streptomycin for approximately 5 days.

To create the shRNA expression vector targeting for human SMUG1 gene, targeted DNA oligonucleotides were sub-cloned into pLKO.1-puro lentiviral vector (Addgene, Watertown, MA, USA). To produce virus particles, HEK293T cells were co-transfected with the plasmids encoding VSV-G lentiviral vector, packaging plasmid pCDNA3-NLbh (Addgene) and the shRNAs. Two days after transfection, the supernatant that contained the virus particles was collected and used for infection of HEK293T cells in the presence of polybrene (8 mg/mL). Continuously expressing SMUG1 shRNA stable cell line was generated in HEK293T cells.

### 2.5. T7E1 Assay

We performed a T7E1 assay to validate the gene modification in SMUG1 KO cell lines. After we extracted the genomic DNA from SMUG1 KO cell lines, we amplified the CRIPSR/Cas9 target sequence in the SMUG1 gene using target sequence-specific primer (F: 5′ TCC-ATC-CAT-GAG-CCT-GCA-GGT-GCC-C 3′, R: 5′ AAG-GAT-GGG-GCT-GGC-GGA-AGG-AAG-A 3′). The amplified products were denatured at 95 °C for 5 min and annealed in order to allow the formation of a heteroduplex between amplified products with and without mutations. After annealing, the products reacted with the T7E1 enzyme at 37 °C for 15 min. We stopped the reaction by adding 1 µL of 0.5 M EDTA and detected the cleaved mismatch products by using agarose gel electrophoresis.

### 2.6. Wound Healing Assay

To determine the effects of cell migration, we seeded approximately 6 × 10^5^ cells/well in 24-well culture plates. A straight scratch that simulated a wound was generated by a 100p pipette tip. After scratching, we removed the suspended cells in the medium and washed them with 1 × PBS twice. The migration of both WT and KD cells were photographed at each indicated time.

### 2.7. RNA-Seq Library Preparation and Sequencing

The RNA-seq library construction was performed using SENSE 3′ mRNA-Seq Library Preparation Kit (Lexogen Inc., Vienna, Austria) according to the manufacturer’s instructions. In brief, each 100 ng total RNA from the cells was prepared, an oligo-dT primer containing an Illumina-compatible sequence at its 5′ end was hybridized to the RNA, and reverse transcription was performed. After degradation of the RNA template, second-strand synthesis was initiated by a random primer containing an Illumina’s compatible linker sequence at its 5′ end. The double-stranded library was purified by using magnetic beads to remove all reaction components. The library was amplified to add the complete adapter sequences required for cluster generation. The finished library was purified from PCR components. High-throughput sequencing was performed as single-end 75 sequencing using Next Seq 500 (Illumina, Inc., San Diego, CA, USA).

### 2.8. Bioinformatical Analysis

The entire analysis pipeline of RNA-seq was coded by using R (ver. 3.6) which was controlled by systemPipeR (ver. 1.18.2). Raw sequence reads were trimmed for adaptor sequence and masked for low-quality sequences using systemPipeR. Transcript quantification of RNA-seq reads was performed with GenomicAlignments (ver.1.20.1) by reads aligned to Ensemble v95 Homo Sapiens transcriptome annotation (GRCh.38.95) using Rsubread (ver. 1.24.6). The FPKM values were calculated using the “fpkm” function from DESeq2 (ver. 1.24.0) that processed them using the robust median ratio method, and transcript reads were normalized by the “voom” function from Limma (ver. 3.40.6). To analyze a transcript as differentially expressed, EdgeR (ver. 3.26.7) calculated the results based on the normalized counts from entire sequence alignments. Significantly differentially expressed transcripts having greater than the fold change of raw FPKM value > 2 and *p* value < 0.01 cases in all experimental comparisons were selected and used for further analysis. Gene annotation was added by the online database using Ensemble biomaRt (ver. 2.40.4), and visualization was performed by using R base code and gplots (ver. 3.0.1.1).

### 2.9. Statistical Analyses

Values are presented as mean ± SEM of experiments performed in triplicate. Data were analyzed by two-way ANOVA followed by Tukey’s multiple comparison test using GraphPad Prism software version 5.01 (San Diego, CA, USA). Differences between groups were considered to be significant at *p* < 0.05.

## 3. Results

### 3.1. Ablation of SMUG1 Induces Apoptotic Cell Death in Hepatocarcinoma HepG2 Cells

Deficiencies in DNA repair lead to the accumulation of DNA damage and the development of cancer and premature aging. In a previous study, we found that *Smug1* was downregulated in ultraviolet B (UVB)-irradiated mouse retinas compared with control retinas [13]. To test the physiological functions of *SMUG1*, we generated SMUG1 KO cells using the CRISPR/Cas9 genome engineering system.

SMUG1-specific sgRNA targeting the fourth exon of the human SMUG1 gene (Figure 1A) was designed using a web-based RNA prediction program (http://crispr.mit.edu). The genomic DNA extracted from the single-cloned cells was used as the template for amplifying the genomic region, including the target site. A T7 endonuclease 1 (T7E1) assay showed that the SMUG1 target site was mismatched by insertion or deletion (Figure 1B). Sanger DNA sequencing of the target region revealed four nucleotide deletions in SMUG1 knock-out cells (Figure 1A). To confirm the silence of SMUG1 expression in the KO cells, SMUG1 protein expression was evaluated by immunoblotting and immunostaining. SMUG1 protein expression was not detected in KO cells (Figure 1C,D). We also generated a SMUG1 knock-down (KD) cell line using shRNA-targeting SMUG1 mRNA (Figure 1E). The KD efficiency was assessed by immunoblotting and immunostaining (Figure 1F,G). These data suggested that the *SMUG1* KO and KD cell lines were successfully established.

### 3.2. Ablation of SMUG1 Inhibits Cell Proliferation and Induces Apoptosis

The cellular phenotypes of *SMUG1* KO cells were assessed by a proliferation assay and an apoptosis assay. The proliferation of WT and *SMUG1* KO cells was assessed by the MTT assay at 12, 24, 36, and 48 h. The viability of *SMUG1* KO cells was significantly decreased after 12 h compared with HepG2 WT cells (Figure 2A). *SMUG1* KD cells also exhibited a reduced proliferation ratio (Figure 2B), and wound healing ability also significantly decreased (Figure 2C).

Cell cycle checkpoints protect the integrity of the cellular genome, which delay cell cycle progression in response to replication stress or diverse types of DNA damage [14]. To verify the relation of cell proliferation inhibition and cell cycle arrest, we analyzed the cell cycle distribution in *SMUG1* KO cells. Cell cycle distribution analysis showed that the proportion of arrested cells at the G2/M phase was significantly increased in *SMUG1* KO cells (32.0% of all counted cells) compared with WT cells (18.8%) (Figure 3A). In addition, the percentage of cells in S and G2 phases was increased in G1-S phase synchronized *SMUG1* KO cells (S: 30.6% and G2: 49.1%) compared with WT cells (S: 10.9% and G2: 23.1%) (Figure 3B, Appendix A). In *SMUG1* KD cells, the cell population ratio of the G2/M phase (21.3%) was higher than that of mock-transfected control cells (18.6%) (Appendix A).

To further test whether the reduced proliferation of KD and KO cells was caused by apoptosis, flow cytometric analysis with annexin V-FITC/propidium iodide (PI) was performed. The proportions of cells at proapoptotic and apoptotic stages in *SMUG1* KO (15.1% and 15.6%, respectively) were higher than in WT cells (4.5% and 5.4%, respectively) (Figure 4A). The proportion of apoptotic cells in the *SMUG1* KD group increased to 4.2% compared with 2.5% among the mock-transfected controls (Figure 4B). In concordance with the cell cycle and apoptosis analysis, the expression of cell cycle-related proteins, cyclin B1, cyclin D1, and p21, was significantly changed in *SMUG1* KO cells (Figure 5A). In addition, BAX and cleaved caspase 3 expression, which are markers of apoptosis, markedly increased in SMUG1 KO cells (Figure 5B). Interestingly, the expression level of a DNA damage marker, phosphorylated gamma-H2AX, was increased in *SMUG1* KO cells (Figure 5B). These data demonstrate that SMUG1 is a critical factor in the proliferation of HepG2 cells via cell cycle and apoptosis regulation.

### 3.3. UVC Irradiation–Mediated Apoptosis Was Accelerated by Ablation of SMUG1

To test whether SMUG1 has the potential to resist DNA damage, various cellular characteristics, such as cell proliferation, DNA fragmentation, and apoptosis, were tested in *SMUG1* KO and KD cells after exposure to UVC irradiation. Cell proliferation was significantly decreased after UVC irradiation in KD cells (Figure 6A). The proliferation rates of WT and *SMUG1* KD cells did not show differences initially, but proliferation ability rapidly decreased 48 h after UVC irradiation. Fragmented gDNA was clearly detected 6 h after UVC irradiation in WT cells. However, fragmented gDNA was detected earlier—4 h after UVC irradiation—in KD cells (Figure 6B).

Furthermore, UVC-induced apoptosis in *SMUG1* KO cells was quantified via flow cytometry analysis using anti-Annexin V FITC antibodies and PI staining (Figure 7A). The proportions of cells at proapoptotic and apoptotic stages in KO cells (66.1% and 17.8%, respectively) were higher than in WT cells (1.6% and 1.3%, respectively). After UVC irradiation, the proportions of proapoptosis and apoptosis in KO cells (60.8% and 27.0%, respectively) were significantly higher than those in WT cells (2.9% and 3.5%, respectively) (Figure 7A). Therefore, ablation of SMUG1 enhances DNA damage and cellular apoptosis after UVC irradiation.

As a consequence of inefficient or absent DNA repair, DNA damage can lead to mutations and cellular abnormalities [14]. When the abnormal cells contain damaged DNA, the G2 checkpoint prevents cells from entering mitosis, providing an opportunity for repair and stopping the proliferation of damaged cells [15]. We next tested whether the increased apoptosis rate was due to a delayed cell cycle. UVC irradiation and SMUG1 KO led to an accumulation of cells in the G2/M phase (29.2%) compared with WT cells (15.1%), coupled with a concomitant decrease in the proportion of cells in the G1 phase (WT and KO, 60.9% and 37.8%, respectively) (Figure 7B). DNA damage and apoptosis were visualized by immunocytochemistry using anti-phospho-H2A.X and cytochrome c antibodies. Phospho-H2A.X and cytochrome c levels were highly detected in *SMUG1* KO cells after UVC irradiation compared with the expression levels in WT cells (Figure 7C). These results indicate the potential of SMUG1 for sensing DNA damage signaled by the increasing level of phospho-gamma-H2AX and as part of cell cycle regulation in UVC-irradiated cells.

### 3.4. Transcriptomic Profiling Revealed that SMUG1 Regulates Gene Expression via Apoptotic and Cell Cycle Transition Pathways

To investigate whether SMUG1 modulates cell proliferation and apoptosis through the regulation of gene expression, RNA-seq analysis was performed using WT and *SMUG1* KO HepG2 cells. Total RNA was extracted from cells, and sequencing libraries were generated from two independent samples. After filtering the RNA-seq data (FPKM > 1, at least in one group), a comprehensive list of 24,444 transcripts was obtained (Appendix A). Principal component analysis using DESeq revealed transcriptomic dynamics in each biological group (Figure 8A). Application of DESeq with a conservative approach to the RNA-seq data obtained from the WT and KO cells identified 1029 differentially expressed (DE) transcripts (RNA-seq FPKM values having ≥ 2 fold change, ≤0.01 false discovery ratio (FDR)). Overall expression patterns of transcripts were shown by the heat maps between WT and KO groups. Unsupervised hierarchical clustering analysis, based on Pearson’s correlation of averaged and log2 of normalized FPKM values of WT and KO groups, showed a decisive shift in the form of upregulated and downregulated transcripts (Figure 8B). All 1029 DE transcripts (502 upregulated and 527 downregulated transcripts) in *SMUG1* KO groups are listed in Appendix A. Compared with the control, the volcano plots were constructed by integrating both the FDR and fold change of each transcript (FDR ≤ 0.01 and absolute log2 (fold change) ≥ 1), to reveal the general scattering of the transcripts and to filter the differentially expressed transcripts in the KO group (Figure 8C). A Venn diagram showed the number of DE transcripts in KO groups (Figure 8D), and hierarchical clustering analysis showed clear changes in the form of upregulated and downregulated transcripts from DE transcripts (Figure 8E).

To identify the transcriptomic pathways affected by the ablation of *SMUG1*, the subset of DE transcripts that was significantly affected in the KO group was subjected to GO annotation using the DAVID bioinformatics resource (GO; https://david.ncifcrf.gov/). The upregulated and downregulated genes were independently subjected to GO analysis to distinguish them according to their functional roles (based on their expression patterns) and not merely according to their gene names. As shown in Figure 8E, the biological processes that were significantly enriched were mainly involved in the regulation of the cell cycle process and mitotic cell cycle phase transition from downregulated genes. The most DEGs (Differentially Expressed Genes)-affected cellular process from upregulated genes was associated with gene silencing, epigenetic regulation of gene expression, and regulation of the apoptotic signaling pathway.

## 4. Discussion

In mammals, a wide range of chemically altered bases was removed through the base excision repair (BER) pathway, and genome integrity was protected by DNA checkpoints, which initiate DNA repair and delay cell cycle progression to prevent replication and segregation of damaged DNA molecules. The transcriptome analysis showed that gene ontology for the regulation of cell cycle (GO:0051726), regulation of cell cycle process (GO:0010564), and regulation of mitotic cell cycle phase transition (GO:19019990) was down-regulated in *SMUG1* KO cells. Among cell cycle regulation genes, the expression of cyclin-dependent kinase 1 (CDK1), which regulates the G2/M transition and mitotic progression, was markedly decreased (Figure 8F). Moreover, the gene ontology for positive regulation of the intrinsic apoptotic signaling pathway (GO:2001244) and apoptotic mitochondrial changes (GO: 0008637), such as *Beclin 1* (*BECN1*), was up-regulated in *SMUG1* KO cells (Figure 8F). BECN1 interacts with BCL-2, which induces the mitochondrial translocation of BAX and the release of proapoptotic factors. It was shown that SMUG1 regulates gene expression of CDK1 and BECN1, and it inhibits mitotic progression and finally undergoes apoptotic cell death.

The alteration of the cellular genome can produce mutations by base-mispairing during DNA replication or can decrease cell viability. Many previous studies suggest that DNA glycosylases initiate the base excision pathway through the cleavage of the N-glycosidic bond in the DNA backbone, and the biochemical function of DNA glycosylases was essential for repair during DNA replication [16]. Here, we analyzed the proportion of the cell cycle in *SMUG1* WT and *SMUG1* KO HepG2 cells to better understand the role of *SMUG1* in cell growth and cell division. We observed that the absence of *SMUG1* induces accumulation of cells in both S and G2-M phases and that it also promotes apoptosis. Ionizing radiation-induced DNA damage is processed by the BER pathway and BER enzymes throughout the cell cycle to investigate the cell cycle-specific repair [17]. We have previously shown that the expression of *SMUG1* was reduced in mouse retina after UV irradiation [13]. In UVC irradiated *SMUG1* KO cells, the cell population in the G2/M phase and undergoing apoptosis is highly increased (Figure 3). In addition, enrichment of H2AX occurred more frequently in the accessible chromatin in *SMUG1* KO cells. Even though we performed the experiments with various aspects, the exact molecular mechanisms of apoptotic cell death in *SMUG1* KO cells needs to be further elucidated. It may involve the DNA repair or replication regulation pathway. Taken together, deficiency of *SMUG1* induces an unstable initial response to replication stress, and it ultimately promotes apoptotic cell death.

## Figures and Tables

**Figure 1 genes-12-00201-f001:**
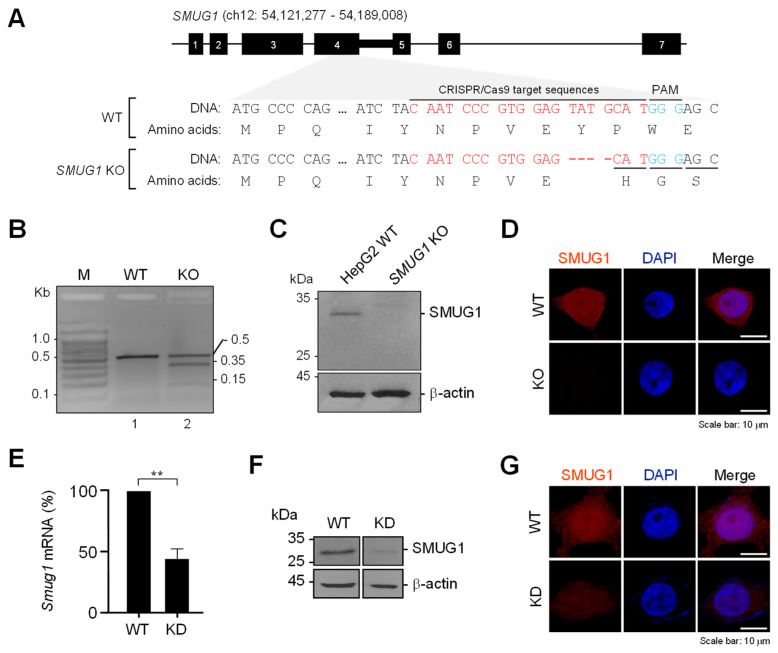
The construction of *SMUG1* knock-out and knock-down hepatocarcinoma HepG2 cells. (**A**) Schematic diagram of the *SMUG1* target sites. sgRNA-targeting sequence was located on the antisense strand of exon 4 of *SMUG1*. The sgRNA-targeting sequence is in red, and the protospacer adjacent motif (PAM) sequence is in blue. (**B**) Validation of *SMUG1* KO for CRISPR/Cas9 efficiency using T7E1 assay. (**C**) Knock-out of *SMUG1* was confirmed by immunoblotting using an anti-SMUG1 antibody. -actin was used as an internal control. (**D**) Knock-out of SMUG1 was visualized by immunocytochemistry using an anti-SMUG1 antibody (red). The cell nuclei were stained with DAPI (blue). (**E**) Validation of SMUG1 mRNA expression by RT-qPCR. (**F**) Knock-down of *SMUG1* was confirmed by immunoblotting using an anti-SMUG1 antibody. β-actin was used as an internal control. (**G**) Knock-down of SMUG1 was visualized by immunocytochemistry using an anti-SMUG1 antibody (red). The cell nuclei were stained with DAPI (blue). Error bars show mean ± SEM. *p* value obtained by Student’s *t*-test. ** *p* value < 0.01.

**Figure 2 genes-12-00201-f002:**
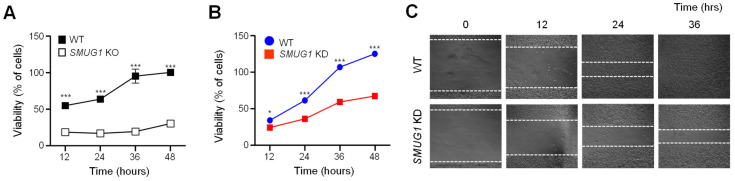
Ablation of *SMUG1* reduces the cell viability and proliferation ratio in hepatocarcinoma HepG2 cells. (**A**) Cell viability was measured by MTT assay from HepG2 WT and *SMUG1* KO cells. (**B**) Cell proliferation was measured by MTT assay from HepG2 WT and *SMUG1* KD cells. (**C**) Wound healing assay was performed in HepG2 WT and *SMUG1* KO cells. Error bars show mean ± SEM*. p* values obtained by Student’s *t*-test. *** *p* value < 0.001, * *p* value < 0.05.

**Figure 3 genes-12-00201-f003:**
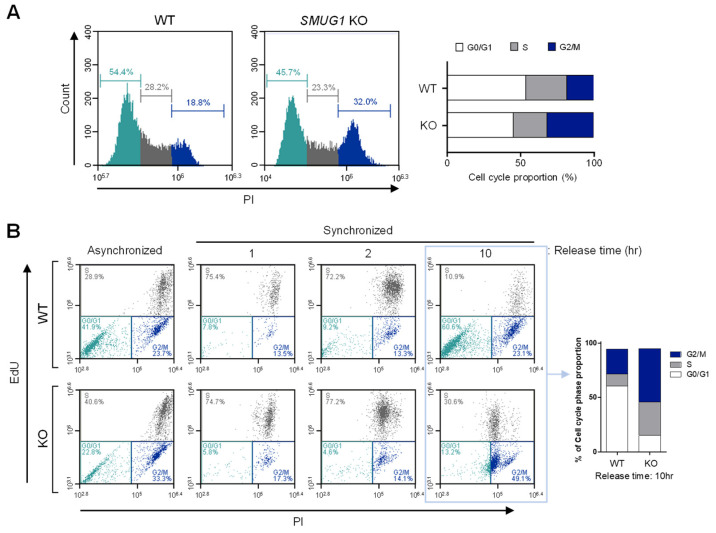
Ablation of *SMUG1* reduces the proliferation through the cell cycle arrest at the G2/M phase. (**A**) Cell cycle assay was performed by the staining of cells with propidium iodide (PI) of asynchronized total cells. Cell cycle proportion was graphed at the right panel. The proportion of the cell cycle was assessed by flow cytometry. The scatter plots represent PI (X-axis) vs. counted cells (Y-axis). (**B**) Cell cycle of HepG2 WT and *SMUG1* KO cells was synchronized using thymidine double blocking. After full synchronization of the cell cycle, cells were released from G1/S arrest with the EdU staining for the newly synthesized DNA. The proportion of the cell cycle was assessed by flow cytometry. The scatter plots represent PI (X-axis) vs. EdU stained cells (Y-axis).

**Figure 4 genes-12-00201-f004:**
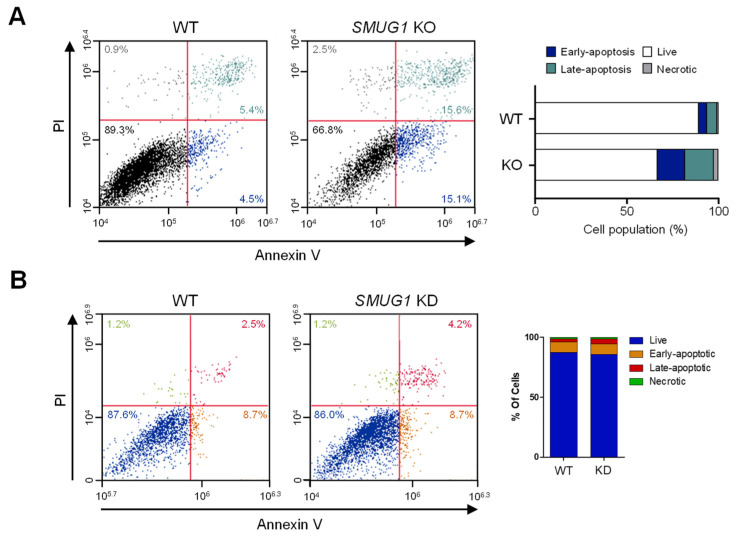
Ablation of *SMUG1* induces apoptotic cell death in *SMUG1* KO and KD cells. (**A**) Apoptotic cell death of HepG2 WT and *SMUG1* KO cells was counted by the flow cytometry analysis. Cells were stained with Annexin V-FITC and propidium iodide (PI). Representative scatter plots of PI (Y-axis) vs. Annexin V-FITC (X-axis). Values are the percentage of cells in the live, early, and late apoptosis and necrosis. (**B**) Apoptotic cell death of HepG2 WT and *SMUG1* KD cells was counted by the flow cytometry analysis. Cells were stained with Annexin V-FITC and propidium iodide (PI). Representative scatter plots of PI (Y-axis) vs. Annexin V-FITC (X-axis). Values are the percentage of cells in the live, early, and late apoptosis and necrosis.

**Figure 5 genes-12-00201-f005:**
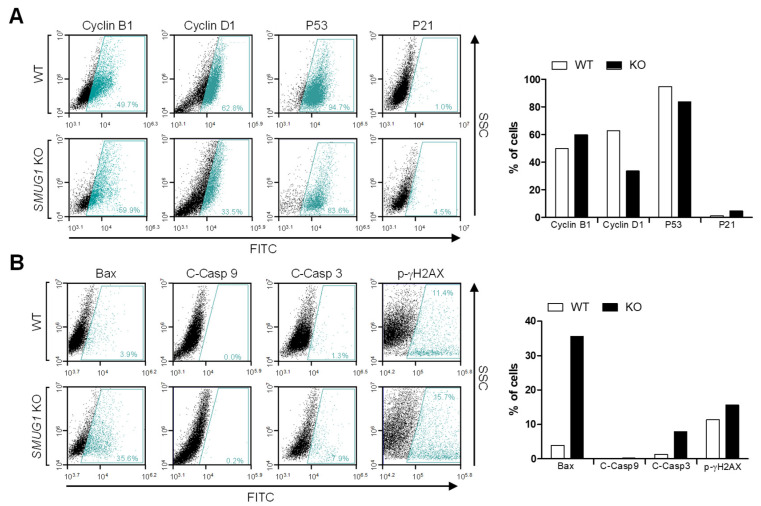
*SMUG1* knock-out inhibits cell proliferation and induces apoptosis. (**A**) Flow cytometry using Cyclin B1, Cyclin D1, P53, and P21 antibodies for the cell cycle and apoptosis analysis of HepG2 WT and *SMUG1* KO cells was performed. (**B**) Flow cytometry using BAX, cleaved caspase 9 (active form), and cleaved caspase 3 (active form) staining for active apoptosis and phosphorylated H2AX for DNA damage response of HepG2 WT and *SMUG1* KO cells was performed.

**Figure 6 genes-12-00201-f006:**
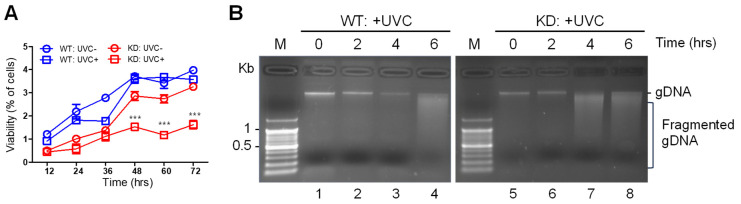
*SMUG1* ablation reduced cell viability and induced gDNA fragmentation after UVC irradiation. (**A**) Cell viability of HepG2 WT and *SMUG1* KD cells irradiated to UVC of 125 mJ/cm^2^ and analyzed by MTT assay. (**B**) Fragmented gDNA was visualized by the agarose gel running after UVC irradiation in SMUG1 WT and KD HepG2 cells. Error bars show mean ± SEM. *p* values obtained by two-way ANOVA. *** *p* value < 0.001; n.s, not significant.

**Figure 7 genes-12-00201-f007:**
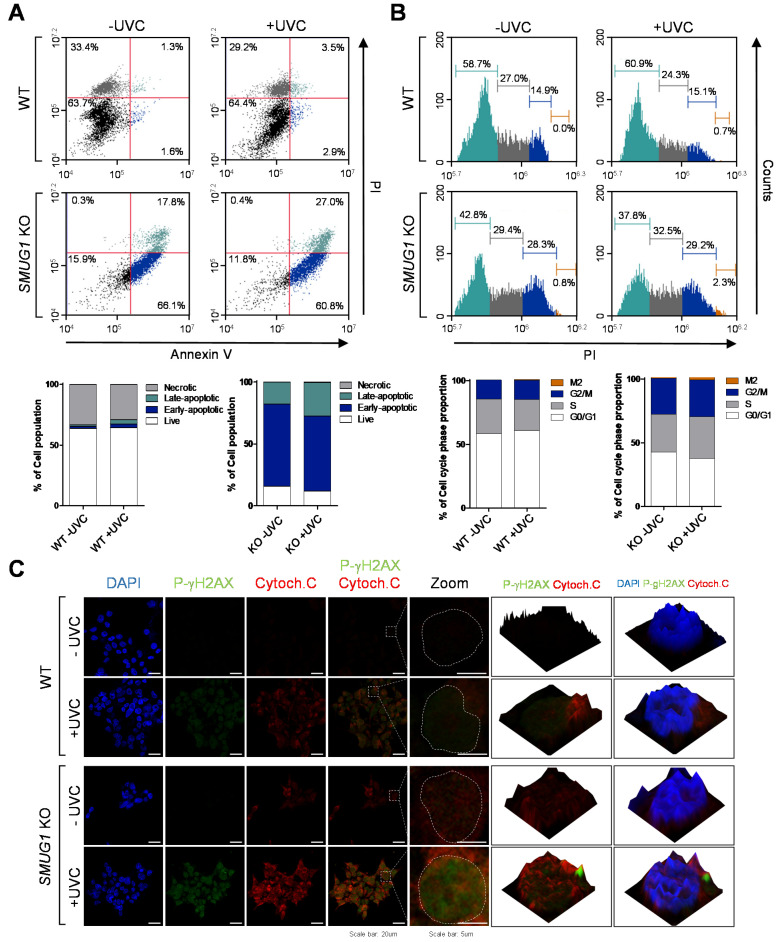
*SMUG1* ablation accelerated UVC irradiation–mediated apoptosis. (**A**) Apoptosis of HepG2 WT and *SMUG1* KO cells irradiated to UVC of 125 mJ/cm^2^ and incubated for 24 h was tested by flow cytometry. Cells were stained with Annexin V-FITC and propidium iodide (PI). Representative scatter plots of PI (Y-axis) vs. Annexin V-FITC (X-axis). Values are the percentage of cells in the live, early, and late apoptosis and necrosis. (**B**) Flow cytometry analysis of cell cycle subversion in UVC-irradiated or non-irradiated WT and KO cells. Cells were fixed in ethanol at 24 h post-infection and stained with PI and applied to flow cytometry. (**C**) HepG2 WT and *SMUG1* KO cells were irradiated with UVC and immunostained with anti-phospho-gamma-H2AX (green) and cytochrome c (red) antibodies. The cell nuclei were stained with DAPI (blue).

**Figure 8 genes-12-00201-f008:**
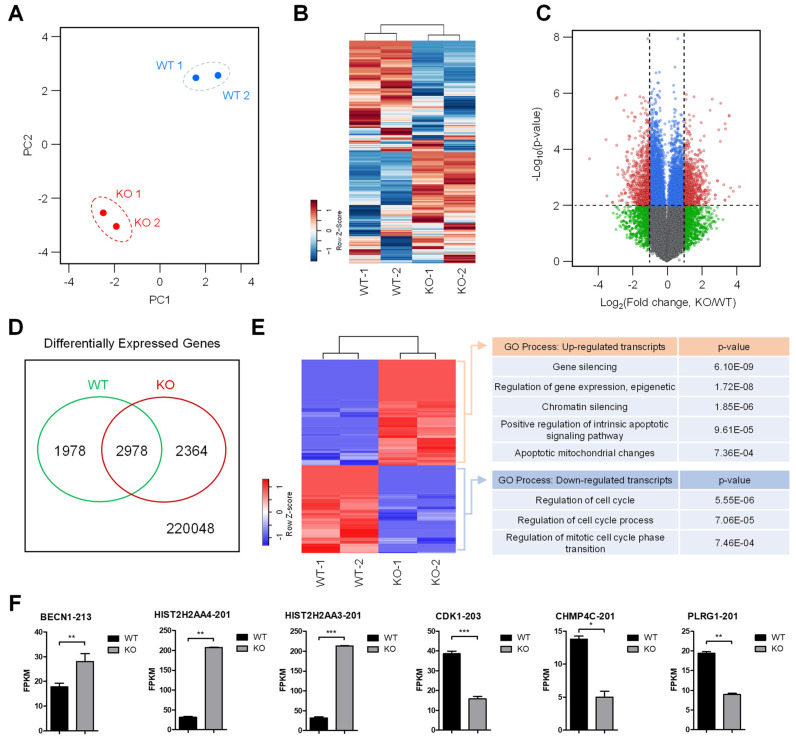
RNA-seq profiles revealed that SMUG1 involved gene expression regulation including cell cycle transition and apoptotic signaling pathways. (**A**) Principal-component analysis (PCA) of directional RNA-seq data. The values reveal the amount of variation attributed to each principal component. Small circles indicate individual samples, and larger show each group. (**B**) Transcriptional pattern analysis in WT and KO groups by employing heat map and hierarchical clustering. (**C**) Volcano plot shows differentially expressed transcripts in the SMUG1 KO group compared to the WT group. Small red-circled transcripts were verified as having the absolute value of log2(fold changes) and −log10(*p* value). (**D**) Differentially expressed transcripts were identified by the DEseq in the SMUG1 KO group from total annotated transcripts. (**E**) Heat map depicting fold changes for all transcripts indicating statistically significant differences between the WT and SMUG1 KO group. The classification of gene ontology (GO) functional enrichment analyses with the DEGs (differentially expressed genes) from comparisons of WT and KO groups. (**F**) Gene expressions of *BECN1*, *HIST2H2AA4*, *HIST2H2AA3*, *CDK1*, *CHMP4C*, and *PLRG1* were designated by the FPKM values of RNA-seq data between SMUG1 WT and KO groups. Error bars show mean ± SEM. *p* values obtained by Student’s *t*-test. *** *p* value < 0.001, ** *p* value < 0.01, * *p* value < 0.1.

## Data Availability

RNA-sequencing data are available for review with GSE145210.

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
