# Peer review of "Ablation of SMUG1 Reduces Cell Viability and Increases UVC-Mediated Apoptosis in Hepatocarcinoma HepG2 Cells"

_genes, 2021, doi:10.3390/genes12020201_

Round 1
Reviewer 1 Report
In this manuscript, the authors reported that the ablation of SMUG1 inhibited cell proliferation and induced cell apoptosis using both KO and KD cell models. Also, they found that the ablation of SMUG1 promoted UVC-mediated apoptosis. Furthermore, using RNA-seq, the authors discovered that the deficiency of SMUG1 led to the downregulation of biological process that were mainly involved in the regulation of cell cycle process and mitotic cell cycle phase transition as well as the upregulation of cellular process that were associated with gene silencing, epigenetic regulation of gene expression, and the regulation of apoptotic signaling pathway. Though the authors addressed interesting questions related to the functional roles of SMUG1, several issues need to be addressed to improve the clarity of this manuscript as well as to consolidate the conclusions.
- Line 63, throughout the text, the SMUG1 KO and KD cell models were generated from the HepG2 cells only. It’s not clear why the “HEK293T” cells were listed here. This needs to be clarified.
- Line 162, “T7E1 assay” was employed to validate the SMUG1 KO using CRISPR/Cas9. However, this method was not described neither in the “Materials and Methods” nor in the “Supplementary Methods”. This must be addressed.
- Throughout the text, it’s not clear how many replicates were repeated for each experiment. Moreover, it seems that no statistical analysis was performed to address whether the observed differences were “statistically significant” or not, such as Fig.1E, Fig. 2A, Fig. 2B, Fig.5, Fig. 6A and Fig. 8F. These must be clarified.
- Fig. 1D and 1E, it’s not clear under what magnification these images were taken. No scale bars were provided either. These must be addressed.
- Fig. 2A and 2B, the Y-axis was illustrated as two different parameters. For better comparison, combine these two figures into one in the context of the same Y-axis.
- Line 194, “wound healing assay” was performed. Again, no description was presented. This must be addressed.
- Fig. 4B, it seems that the scale of PI staining was not right. This needs to be confirmed.
- Fig. 6B, it’s not clear how much gDNA were loaded for each sample for the “DNA fragmentation assay”. To compare the “fragmented gDNA” among different samples, the input amount should be the same. However, it is obvious that “lane 3” contained less DNA compared with all other lanes. This must be clarified.
- Fig. 7A, “Early-apoptotic” was incomplete in the annotations of the bottom panels.
- Fig. 7C, “Merge” should be done for “DAP1, P-γH2AX and cytochrome c” other than “P-γH2AX and cytochrome c” only. Again, it’s not clear under what magnification these images were taken. No scale bars were provided. All these must be addressed.
- Lines 272-276, the description of the results were not matched with the data presented in the Fig. 7A. This must be clarified.
- Fig. 8B, the labeling for the samples was missing.
- Lane 371, the results described here should be corresponding to the data presented in the Fig. 7 other than Fig. 3.
- This manuscript entitled “Ablation of SMUG1 accelerates UVC-mediated apoptosis via downregulation of DNA damage response pathways”. However, for the RNA-seq analysis, the authors only made comparisons between SMUG1 KO and WT cells without the exposure to UVC. In order to consolidate the conclusions the title conveyed, the authors should also perform the RNA-seq analysis for SMUG1 KO and WT cells under the challenge of UVC.
Reviewer 2 Report
DNA base damage, frequently derived from uracil misincorporation or base oxidation, is the most abundant DNA lesion encountered by cells. Base excision repair (BER) is largely responsible for the replacement of the damaged base. In the BER pathway, the damaged base must be removed by glycosylase first to expose an abasic site for the AP endonuclease to act. For uracil, The human genome encodes several uracil-DNA glycosylases including UNG1, SMUG1, TDG, and MBD4. Among them, UNG1 is well characterized and responsible for the removal of the majority of mis-incorporated uracils in DNA duplex. The rest three are relatively under-studied. In the current manuscript by An et.al, knockout and knockdown cell lines of SMUG1 were created and characterized for their growth in the absence or presence of UV treatment. The SMUG1 knockout cells displayed a slower growth rate with prolonged S and G phases and an increase in the fraction of apoptotic cells, which was further aggravated by the UV treatment. Furthermore, utilizing RNA-seq, the authors classified the genes whose expression is altered by SMUG1 knockout. The data are very clean. The material generated and results collected in this research will build a foundation for further studies on SMUG1, therefore will be of interest to the people in the field. I have a couple of suggestions on the writing for the authors to consider.
- Page 2, lane 48-49, “although SMUG1 induces DNA repair in damaged cells, the functional role of SMUG1 remains unclear.” This sentence laid out the question that the authors were trying to address. However, it is very vague. Please be more specific here.
- UNG1 removes uracil embedded in a duplex, while the activity of SMUG1 is single-strand selective. Note that the deamination of cytosine occurs mostly on single-stranded DNA. The resulted uracil, therefore, may serve as a good substrate for SMUG1. I wonder whether the authors can discuss this.
